# Muscarinic receptor M3 contributes to intestinal stem cell maintenance via EphB/ephrin-B signaling

Toshio Takahashi[1] , Akira Shiraishi[1], Jun Murata[1] , Shin Matsubara[1] , Satsuki Nakaoka[2] , Shinji Kirimoto[2] , Masatake Osawa[3,4]

Acetylcholine (ACh) signaling through activation of nicotinic and muscarinic ACh receptors regulates expression of specific genes that mediate and sustain proliferation, differentiation, and homeostasis in the intestinal crypts. This signaling plays a pivotal role in the regulation of intestinal stem cell function, but the details have not been clarified. Here, we performed experiments using type 3 muscarinic acetylcholine receptor (M3) knockout mice and their intestinal organoids and report that endogenous ACh affects the size of the intestinal stem niche via M3 signaling. RNA sequencing of crypts identified up-regulation of the EphB/ephrin-B signaling pathway. Furthermore, using an MEK inhibitor (U0126), we found that mitogen-activated protein kinase/extracellular signal-regulated kinase (MAPK/ERK) signaling, which is downstream of EphB/ephrin-B signaling, is activated in M3-deficient crypts. Collectively, M3, EphB/ephrin-B, and the MAPK/ERK signaling cascade work together to maintain the homeostasis of intestinal epithelial cell growth and differentiation following modifications of the cholinergic intestinal niche.

## Introduction

The ability of intestinal stem cells (ISCs) to divide and differentiate is necessary for tissue repair and homeostasis. Maintenance of a functional intestine requires appropriate spatial and temporal processes involving multiple key signals from the surrounding niche (Middelhoff et al, 2020; Takahashi & Shiraishi, 2020). The ISC niche in the small intestine is composed of stem cells and Paneth cells, and is surrounded by mesenchymal cells at the crypt bottom (Sato et al, 2011; Tian et al, 2011). This provides a unique microenvironment that constitutes a constantly renewing dynamic system along the crypt-villus axis throughout postnatal life (Lopez-Garcia et al, 2010; Snippert et al, 2010). Marked leucine-rich repeat-containing G-protein coupled receptor 5 (Lgr5)–expressing ISCs temporarily produce undifferentiated cells that divide rapidly while

moving toward the intestinal lumen (Batlle et al, 2002). During migration, these cells differentiate into mature cells such as goblet cells, tuft cells, enteroendocrine cells, and absorptive cells (enterocytes) (Potten & Loeffler, 1990; Stappenbeck et al, 1998; McKinley et al, 2017). Paneth cells move to the crypt bottom. These special mechanisms are important for life-long steady-state maintenance of the epithelium (Sangiorgi & Capecchi, 2008; Voog & Jones, 2010; Takeda et al, 2011; Tian et al, 2011).

Acetylcholine (ACh) and its receptors (nicotinic and muscarinic receptors) are important constituents of the cholinergic system. Muscarinic receptors (M1-M5) are G protein-coupled receptors that mediate mucosal ion transport (Cameron & Perdue, 2007), epithelial proliferation (Gross et al, 2012), barrier function (McLean et al, 2016), and immune host defense mechanisms (Labed et al, 2018), as well as cholinergic neurotransmission at effector cells. One of the receptor subtypes, M3, which is expressed widely throughout the gastrointestinal tract, couples to $G_{\alpha q/11}$ to increase intracellular calcium via activation of phospholipase C signaling and inositol phosphate formation. Thus, M3 signaling alters cell function, including proliferation and differentiation (Slack, 2000; Takahashi et al, 2014). Our previous pharmacological studies with crypt-villus organoids revealed that ACh is synthesized in intestinal epithelial cells and plays a role in cell division and differentiation of Lgr5-positive (Lgr5[+]) ISCs in the small intestine by binding to muscarinic receptors including M3 in vitro (Takahashi et al, 2014). How ISC proliferation, differentiation, and maintenance are controlled and which inductive signals are required for tissue maintenance are well established (Takahashi & Shiraishi, 2020). However, little is known regarding the regulation of these pathways in vivo.

Erythropoietin-producing hepatocellular carcinoma cell (Eph) receptors and their ligands (ephrins) form a complex signaling network that is involved in multiple processes during morphogenesis (Pasquale, 2005; Batlle & Wilkinson, 2012; Klein, 2012). Eph receptors and ephrins are divided into two classes. EphA receptors bind to glycosylphosphatidylinositol moiety-anchored ephrin-As, and EphB receptors bind to transmembrane ephrin-Bs (Gale et al, 1996; Klein, 2012; Lisabeth et al, 2013). Increasing evidence supports

[1]Suntory Foundation for Life Sciences, Bioorganic Research Institute, Kyoto, Japan  [2]KAC Co., Ltd. Kyoto, Japan  [3]Department of Regenerative Medicine and Applied Biomedical Sciences, Graduate School of Medicine, Gifu University, Gifu, Japan  [4]Center for Highly Advanced Integration of Nano and Life Sciences, Gifu University (G-CHAIN), Gifu, Japan

Correspondence: takahashi@sunbor.or.jp

the roles of EphB/ephrin-B signaling in intestinal crypts. Wnt signaling positively regulates cell proliferation by controlling expression of EphB2 and EphB3 (Batlle et al, 2002). EphB2 receptor-positive cells are found in intervillus pockets in the small intestine of neonatal mice and in crypts in the adult small intestine (Batlle et al, 2002). Conversely, Wnt signaling represses expression of ephrin-B1 and ephrin-B2 in differentiated cells, and consequently, the distribution of the ligands is largely complementary to EphB receptors (Batlle et al, 2002). In adult mice, proliferating undifferentiated cells (progenitor cells) are positive for EphB2, and Paneth cells preferentially express EphB3 (Batlle et al, 2002). Double knockout of EphB2 and EphB3 results in complementary expression of EphB receptors and ephrin-B ligands to sustain the normal organization of Lgr5$^+$ ISCs, Paneth cells, and dividing uncommitted cells in the base and differentiated cells in the apex (Batlle et al, 2002; Merlos-Suárez et al, 2011). EphB2 and EphB3 also control division of uncommitted cells and re-entry of quiescent cells into the cell cycle (Holmberg et al, 2006). Collectively, in the presence of an increasing gradient of EphB2 receptors, ephrin-B ligand-positive precursors are organized from top to bottom in crypts according to a reverse gradient of ephrin-B1 and ephrin-B2 expression.

Given the key role of M3 expression in proliferation and differentiation of ISCs, determining whether genetic ablation of M3 alters homeostasis of the functional intestine in vivo is important. Although some studies support the importance of epithelial tuft cells as epithelial niche cells and the importance of intestinal homeostasis via M3 signaling, gaining more insight into stem cell signaling pathways in the small intestine is also important (McLean et al, 2016; Middelhoff et al, 2020). Here, we show that M3 signaling coordinates the EphB/ephrin-B signaling cascade to sustain intestinal epithelial homeostasis, which in part plays a role in regulating the rate of proliferation of epithelial cells.

# Results

## Small intestinal crypt size of M3$^{-/-}$ mice is increased despite the body size

We have reported that endogenous ACh released from the intestinal epithelium maintains homeostasis of intestinal epithelial cell growth and differentiation via M3 (Takahashi et al, 2014). In line with the importance of M3 for intestinal homeostasis, we analyzed small intestinal phenotypes in M3$^{-/-}$ mice compared with that of WT mice. Adult M3$^{-/-}$ mice exhibited reduced longitudinal growth (Fig S1A) (Yamada et al, 2001; Gautam et al, 2006). The body weight of M3$^{-/-}$ mice was also decreased to 71.3% of that of their control littermates (Fig S1B) (controls: 29.33 ± 0.78 g, versus M3$^{-/-}$ 20.9 ± 2.69 g, n = 3 each; $P < 0.05$). On the other hand, the length of the small intestine of M3$^{-/-}$ mice showed no significant difference compared with WT mice (Fig S1C). Individuals of the same genotype can vary substantially in size, whereas the shapes and sizes within and between tissues are precisely maintained. This adaptation of pattern with size is termed "scaling" (Ben-Zvi et al, 2011). Contrary to the scaling theory, the shape and size of the small intestine of M3$^{-/-}$ mice was nearly the same as that of WT mice. We, therefore, focused on analysis of the small intestinal tissue of M3$^{-/-}$ mice in detail.

Histological examination of hematoxylin and eosin (H&E)–stained sections revealed that the crypt depth of M3$^{-/-}$ mice appeared to increase in the duodenum, jejunum, and ileum (Fig 1A). Deletion of M3 appeared to not affect the intestinal crypt-villus architecture regarding villus height (Fig 1A). As shown in Fig 1B and C, crypts isolated from M3$^{-/-}$ duodenum were clearly ~2 times larger in length and area than those isolated from WT duodenum. Thus, the length of the small intestine of M3$^{-/-}$ mice may be due to an increased number of stem/progenitor cells and/or an increased number of crypts per villus in the M3$^{-/-}$ small intestine.

## Deletion of M3 increases rates of proliferation and migration along the crypt-villus axis

In previous in vitro studies, we revealed that treatment with an M3 antagonist results in increased proliferation and differentiation of Lgr5$^+$ stem cells (Takahashi et al, 2014). To confirm the pharmacological effect, we examined the effect of M3 deletion on the rate of proliferation of intestinal epithelial cells in vivo. To determine whether the crypt-villus areas in M3$^{-/-}$ small intestine contained actively proliferating cells, the tissue sections were stained with an antibody against the proliferation marker Ki67, and the number of positive cells in the small intestine was counted. Ki67$^+$ cells were found in the stem cell compartment and the transit-amplifying area (Figs 2A and S2A and B). A significant difference was found between WT and M3$^{-/-}$ mice in the number of Ki67$^+$ cells in the crypts (Fig 2B). At 15 wk of age, mice were pulse-labeled in the S-phase with BrdU and euthanized 3, 6, 15, and 24 h later for immunohistochemical examination. The rate of migration of BrdU-positive (BrdU$^+$) epithelial cells along the crypt-villus axis was higher in the small intestine of M3$^{-/-}$ mice than in control littermates at all times examined (Figs 2C and S2C and D). The number of BrdU$^+$ cells was also higher in the small intestine of M3$^{-/-}$ mice than in control littermates at all times examined (Fig 2D). We concluded that the number of stem and/or progenitor cells was increased by M3 deletion, leading to an increase in the number of BrdU-incorporating cells, which resulted in an increase in the crypt size.

## Deletion of M3 enhances organoid growth and marker gene expression except for chromogranin A

Gene expression analysis was performed to determine whether deletion of the M3 gene alters the gene expression pattern of mRNAs encoding other muscarinic receptor subtypes in cultured organoids. Loss of expression of M3 was confirmed in the M3$^{-/-}$ organoids (Fig 3A). RT–PCR analysis revealed that the gene expression pattern of other subtypes was similar to that of organoids derived from WT crypts (Fig 3A). To validate the RT–PCR data, the transcription level of the genes was examined with quantitative RT–PCR (qRT-PCR). The analysis confirmed that no differences were present between WT organoids and M3$^{-/-}$ organoids in the gene expression pattern of other subtypes (Fig 3B).

Next, we examined the effect of deletion of M3 on organoid growth. Organoid growth was clearly enhanced in M3$^{-/-}$ organoids compared with WT organoids at days 3 and 6 (Fig 3C). The enhancement of organoid growth occurred at day 1 (Fig 3D). At all time points, a statistically significant effect was detected (Fig 3D). To

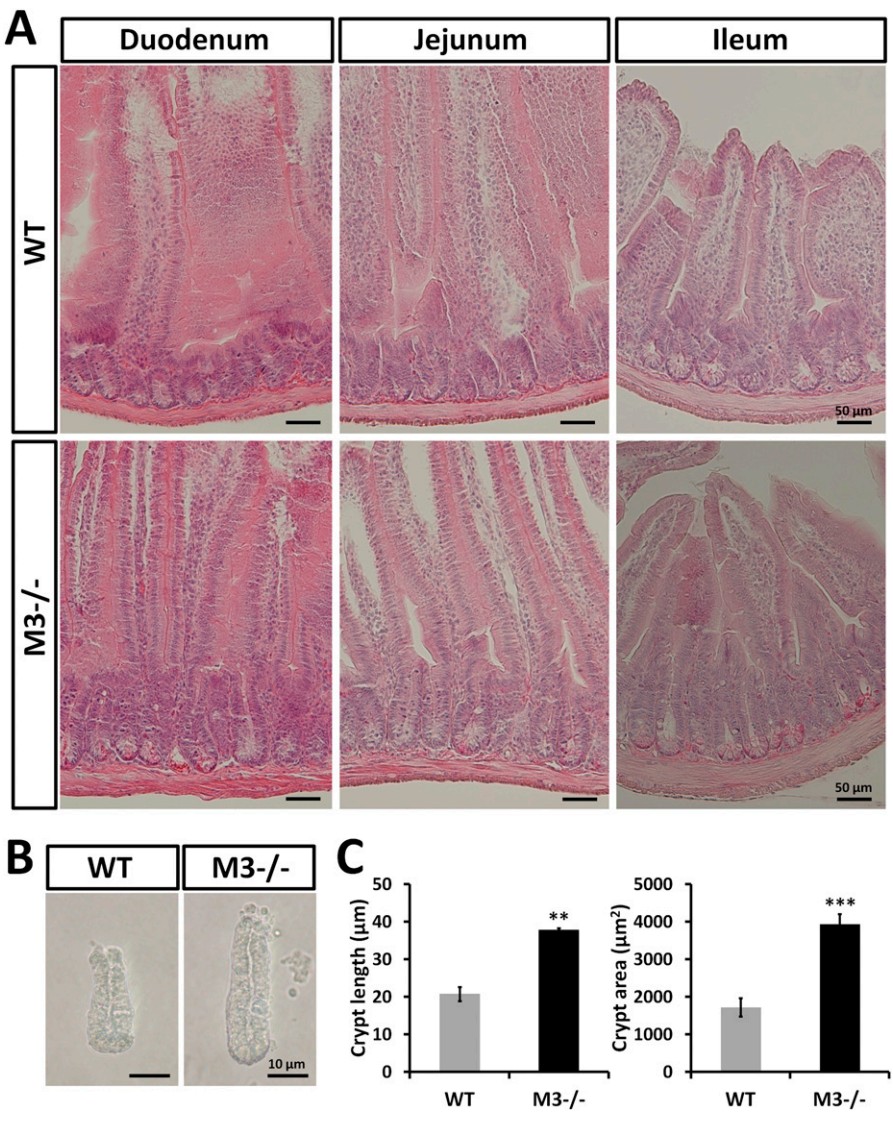

**Figure 1. Genetic ablation of type 3 muscarinic receptor induces an increase in intestinal crypt size.**
**(A)** Histological analysis with hematoxylin and eosin staining of small intestinal regions (duodenum, jejunum, and ileum) of WT and M3$^{-/-}$ mice. **(B)** Representative images of crypts isolated from WT and M3$^{-/-}$ duodenal regions. **(C)** Isolated crypt size was compared between WT and M3$^{-/-}$ mice. Data information: The results are based on three independent experiments and are presented as mean values ± SD. The statistical significance was calculated with $t$ test (**$P$ < 0.005; ***$P$ < 0.0005). Source data are available online for this figure.

further evaluate the functional consequence of deletion of *M3*, we analyzed mRNA expression of selected genes in crypts and epithelium. The ISC marker (*Lgr5*) was significantly enhanced in M3$^{-/-}$ crypts (Fig 3E). mRNA levels of epithelial cell markers for Paneth cells (*lysozyme*), enterocytes (*villin*), and goblet cells (*mucin-2*) were significantly enhanced in M3$^{-/-}$ intestinal epithelium, but the marker for enteroendocrine cells (*chromogranin A*) was not (Fig 3E). No gross expansion of chromogranin A–positive cells, which would possibly indicate the importance of epithelial-stromal crosstalk in endogenous ACh secretion (Middelhoff et al, 2020), was observed.

### Intestinal M3 deletion leads to activation of the EphB/ephrin-B system

Muscarinic receptor signaling pathways govern cell growth and proliferation (Gross et al, 2012). Single-cell RNA-sequencing (RNA-seq) data derived from small intestinal epithelial cells showed that the highest level of M3 expression was present in secretory cells (Paneth and goblet cells), followed by stem and endocrine cell compartments (Haber et al, 2017). Thus, all cells appear to be able to respond to alternations in cholinergic signaling and thus are candidates for cholinergic sensor cells. To gain further insights into functionally active regulatory networks in the crypts, we carried out RNA-seq analysis (SRA accession numbers: SRR12489254–SRR12489266).

The reads from WT and M3$^{-/-}$ crypts were mapped to the mouse genome (mm10), and differentially expressed genes (DEGs) in WT and M3$^{-/-}$ crypts were detected with the $t$ test and a significance level of $P$ < 0.05. Up-regulated and down-regulated DEGs were subjected to enrichment analysis with Kyoto Encyclopedia of Genes and Genomes (KEGG) pathways using the Database for Annotation, Visualization, and Integrated Discovery (DAVID) as previously described (Takahashi et al, 2018). The up-regulated DEGs were enriched in the "TNF signaling pathway" (mmu04668), "MAPK signaling pathway" (mmu04010), "Cell cycle" (mmu04110), and "Toll-like receptor signaling pathway" (mmu04620) with $P$-values of 6.82 × 10$^{-10}$, 9.77 × 10$^{-9}$, 3.39 × 10$^{-7}$, and 4.30 × 10$^{-7}$, respectively (Fig 4A). The "TNF signaling

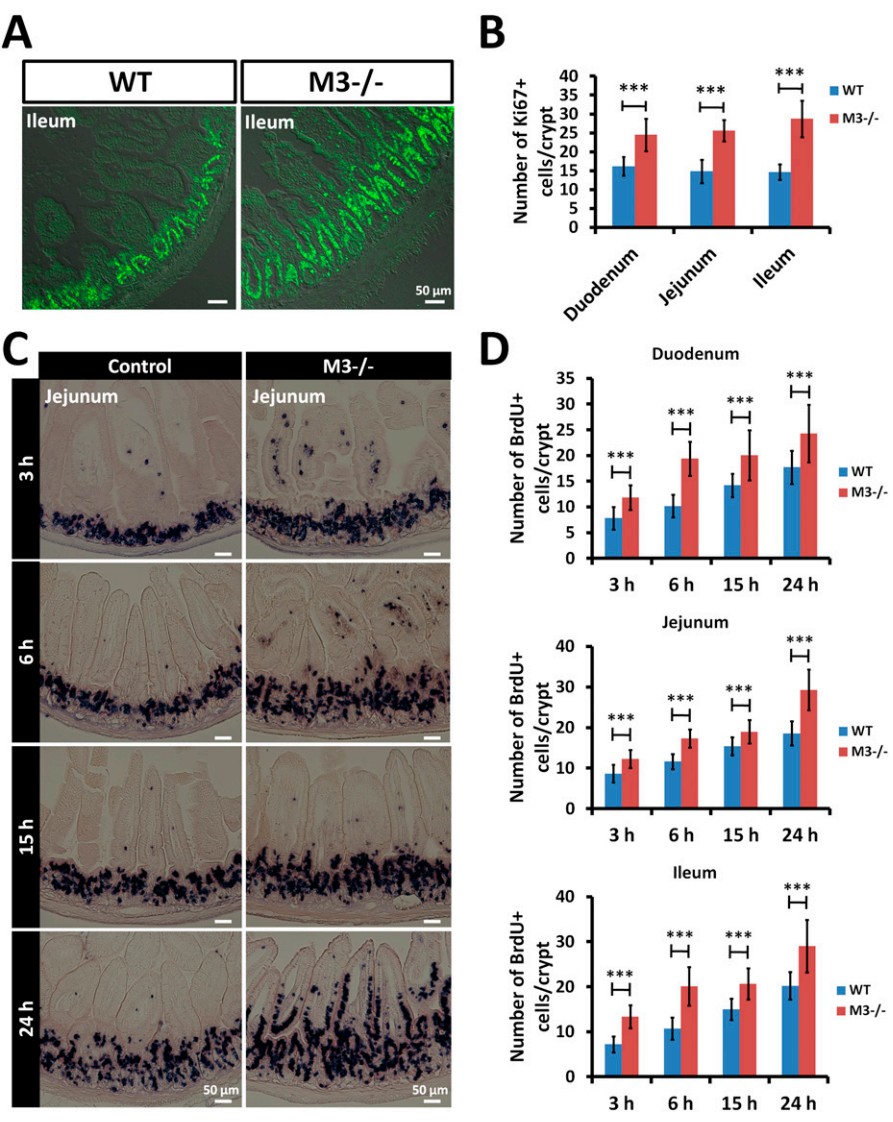

**Figure 2. Deletion of M3 increases the rate of proliferation and migration of small intestinal epithelial cells.**

**(A)** Representative images of anti-Ki67 antibody staining in the ileal regions of WT and M3$^{-/-}$ mice. **(B)** The number of Ki67-positive cells per crypt in the small intestine of WT and M3$^{-/-}$ mice. At least 20 crypts were examined per mouse per genotype (n = 3 mice per genotype). **(C)** Control and M3$^{-/-}$ mice were injected with BrdU intraperitoneally and euthanized 3, 6, 15, and 24 h later. Representative sections of jejunal regions from control and M3$^{-/-}$ mice at 3, 6, 15, and 24 h. **(D)** The total number of BrdU-positive cells per crypt in the small intestine was compared between control and M3$^{-/-}$ mice at 3, 6, 15, and 24 h. At least 30 crypts were examined per mouse per genotype per time point (n = 3 mice per genotype per time point). Data information: The results are based on three independent experiments and are presented as mean values ± SD. The statistical significance was calculated with *t* test (***$P$ < 0.0005).

Source data are available online for this figure.

pathway" genes were up-regulated in the "MAPK signaling pathway," including *ASK1, Tak1, Tpl2, Nik, MKK4/7, MKK3/6, ERK1/2, JNK1/2,* and *p38,* but not genes upstream of *Tnfr1* and *Tnf* (Fig S3A and B). The up-regulated DEGs for the "MAPK signaling pathway" included the same genes in the "TNF signaling pathway" (Fig S4A and B). The genes downstream of the "MAPK" signaling pathway that were up-regulated included genes in the "Cell cycle" pathway. The up-regulated DEGs for "Cell cycle" included *CDK4,6, CycA, Mad2, Cdc7, Cdc45, Cdc25a, CDC25B/C,* and *APC/C,* suggesting that the "Cell cycle" genes were up-regulated via the MAPK signaling pathway (Fig S5A and B). Similar to the "TNF signaling pathway," DEGs in the "Toll-like receptor signaling pathway" also included the "MAPK signaling pathway" genes, but not genes upstream of *Cd14* and *Tlr4* (Fig S6A and B). CD14 has been shown to be required for lipopolysaccharide-induced TLR4 endocytosis (Zanoni et al, 2011).

The down-regulated DEGs were enriched in "Linoleic acid metabolism" (mmu00591) (Figs 4A and S7A and B), "Steroid hormone biosynthesis" (mmu00140) (Figs 4A and S8A and B), "Retinol metabolism" (mmu00830) (Figs 4A and S9A and B), and "Chemical carcinogenesis" (mmu05204) (Figs 4A and S10A and B) with *P*-values of 2.30 × 10$^{-8}$, 6.80 × 10$^{-7}$, 1.80 × 10$^{-5}$, and 2.10 × 10$^{-5}$, respectively (Fig 4A). All down-regulated DEGs involved in these four pathways were P450 genes (*Cyp2c40, Cyp2c55, Cyp2d26, Cyp2e1, CYP2J, Cyp2s1, Cyp3a11, Cyp3a13,* and *Cyp3a25*) (Figs S7–S10A and B). "Chemical carcinogenesis"–related P450 down-regulation coincides with previously reported cancer suppression in M3$^{-/-}$ mice (Raufman et al, 2008, 2011).

Of interest, DEGs in "Axon guidance" (mmu04360) highlighted the up-regulation of both EphB2 and ephrin-B2 (Fig 4B), which both regulates ERK and *PAK* (Fig 4B and C). Eph receptors and ephrins are involved in cell positioning along the crypt-villus axis as well as axon guidance (Batlle et al, 2002; Palmer & Klein, 2003). These enriched pathways suggest that the increase in the intestinal crypt size in M3$^{-/-}$ mice is caused by cell cycle promotion of EphB/ephrin-B signaling via MAPK modules.

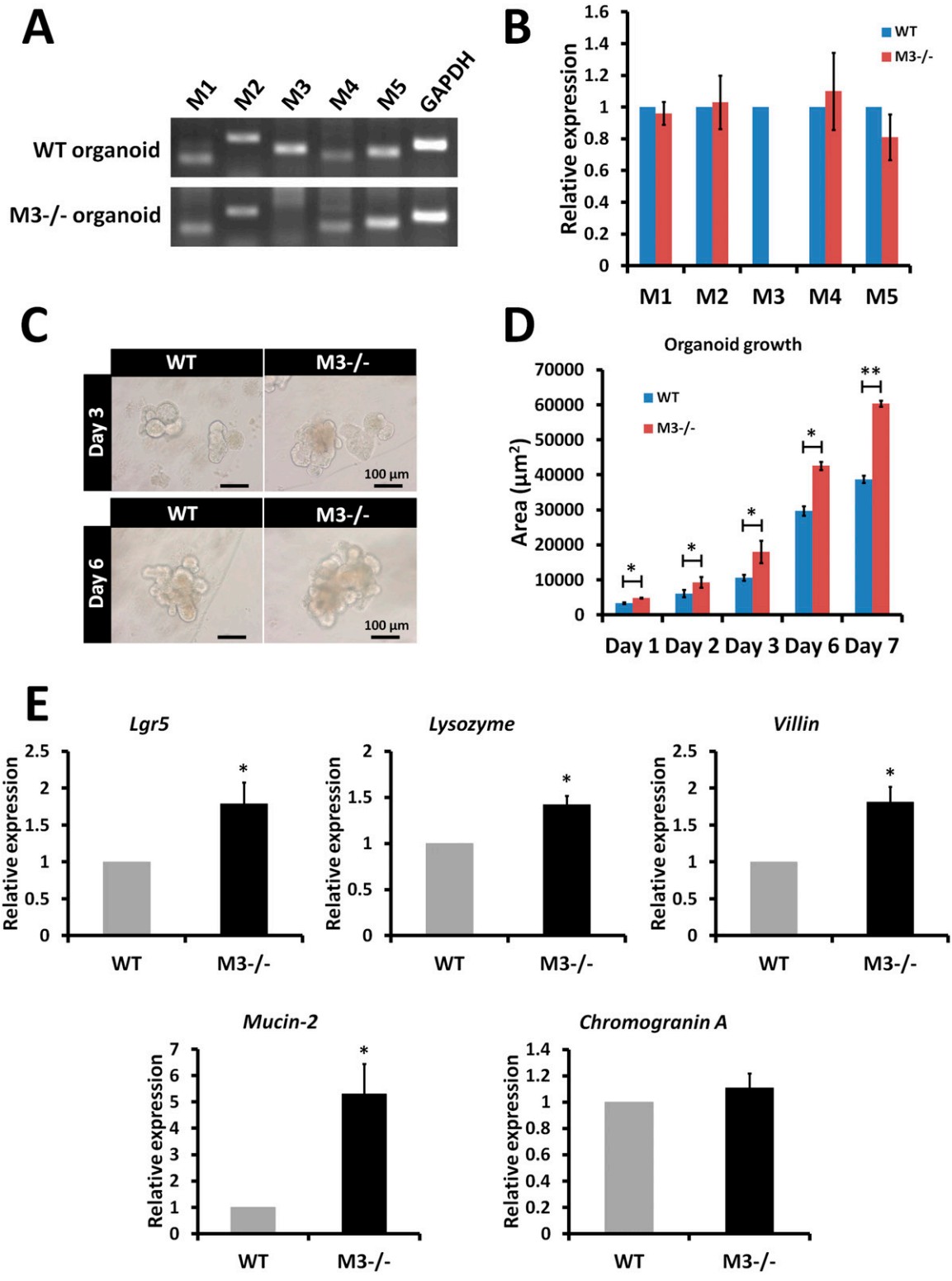

**Figure 3. Comparison of expression of muscarinic receptors and marker genes in the organoids, crypts, and epithelium of M3$^{-/-}$ and WT mice.**
**(A)** Expression of muscarinic receptors (*M1, M2, M3, M4,* and *M5*) in 3-d cultured WT and M3$^{-/-}$ organoids. **(B)** Relative quantification of muscarinic receptor genes in 3-d cultured M3$^{-/-}$ organoids. **(C)** Micrographs of organoids in 3-and 6-day cultured WT and M3$^{-/-}$ organoids. **(D)** Comparison of organoid growth from crypts in WT and M3$^{-/-}$ mice. **(E)** Relative quantification of the marker genes (*Lgr5, Lysozyme, Villin, Mucin-2,* and *Chromogranin A*) in M3$^{-/-}$ crypts and epithelium. Data information: The results are based on three independent experiments and are presented as mean values ± SD. The statistical significance was calculated with *t* test (*$P < 0.05$; **$P < 0.005$). *GAPDH* was used as the internal control.
Source data are available online for this figure.

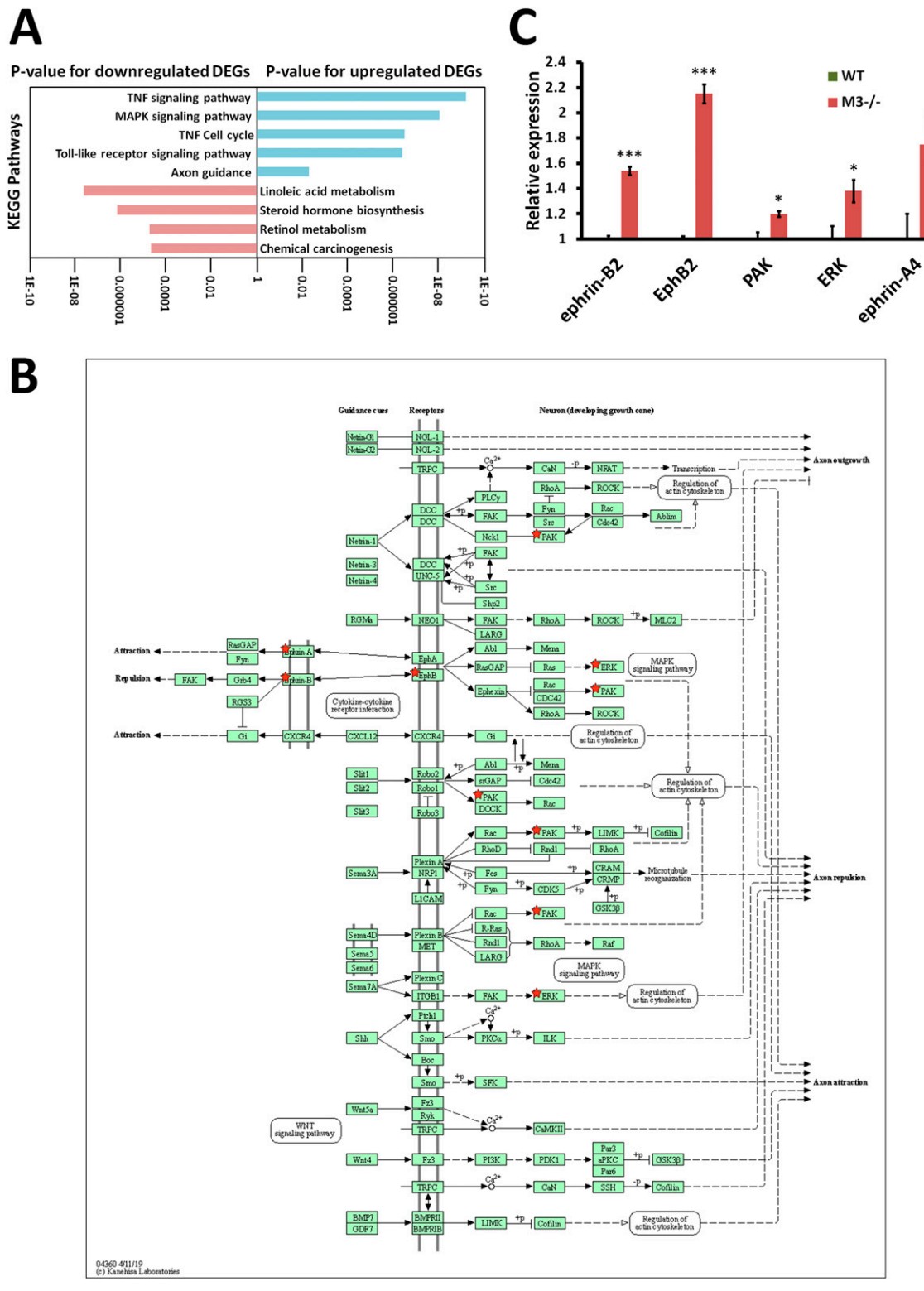

**Figure 4. RNA-seq analysis indicates EphB/ephrin-B signaling pathway activation in M3$^{-/-}$ mice.**
**(A)** Database for Annotation, Visualization, and Integrated Discovery analysis of genes differentially expressed in the knockout mice shows a gain of characteristics of proliferation and differentiation and down-regulation of four pathways associated with metabolism. **(B)** EphB/ephrin-B signaling pathway (mmu04360) maps derived from Database for Annotation, Visualization, and Integrated Discovery analysis. Green boxes and arrows indicate the genes and interactions in the pathway. +p and −p denote phosphorylation and dephosphorylation, respectively. The genes with red stars are those that were up-regulated in M3$^{-/-}$ crypts. **(C)** Differential expression patterns of *ephrin-B2*, *EphB2*, *PAK*, *ERK*, and *ephrin-A*4 in M3$^{-/-}$ crypts. Data information: The results of RNA-seq are based on three independent experiments and are presented as mean values ± SD. The statistical significance was calculated with *t* test (*$P < 0.05$ and ***$P < 0.0005$).
Source data are available online for this figure.

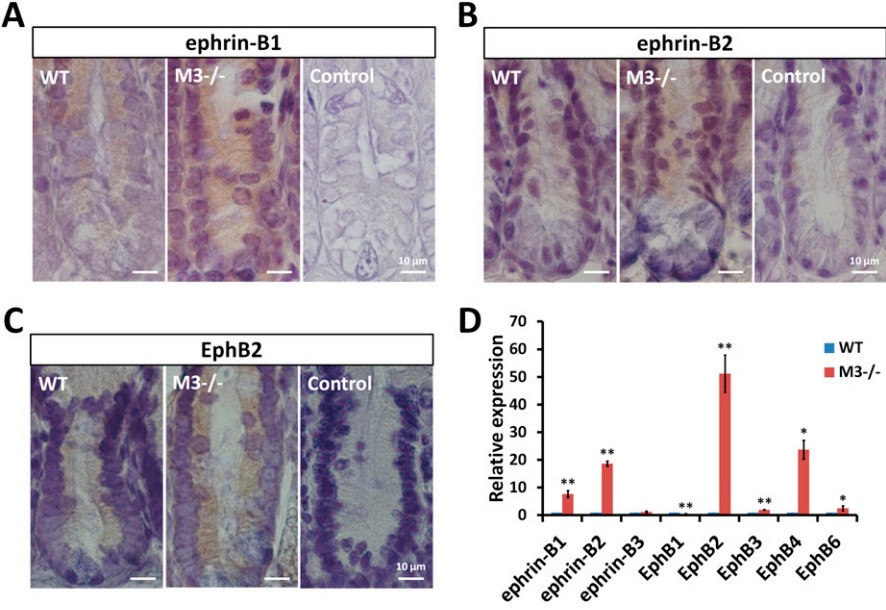

**Figure 5. Expression pattern of EphB/ephrin-B molecules in the adult small intestine.**
**(A, B, C)** Immunodetection of ephrin-B1, ephrin-B2, and EphB2 proteins in WT and M3$^{-/-}$ crypts. Control sections were labeled with secondary antibody in the absence of primary antibodies. **(D)** Relative quantification of EphB/ephrin-B molecules in M3$^{-/-}$ crypts. Data information: The results of qRT-PCR are based on three independent experiments and are presented as mean values ± SD. The statistical significance was calculated with $t$ test (*$P < 0.05$ and **$P < 0.005$).
Source data are available online for this figure.

## M3 is involved in regulation of the expression pattern of the EphB/ephrin-B system

The overall structure of the adult small intestine is more complex than that of the neonatal intestine (Montgomery et al, 1999; Spence et al, 2011; Takahashi, 2020). ISCs present at the crypt bottom give rise to a transient population of proliferating cells. These cells divide rapidly while migrating toward the intestinal lumen. Goblet cells, enteroendocrine cells, and enterocytes migrate toward the villus. Paneth cells follow a downward migration path. Paneth cells interact with ISCs via production of bactericidal products and signaling molecules (Takahashi & Shiraishi, 2020).

A complex gradient of factors maintains ISC stemness and proliferation along the crypt-villus axis. The expression pattern of the EphB/ephrin-B system reflects this environment (Batlle et al, 2002). Thus, to explore in more detail whether the EphB/ephrin-B system is altered in M3$^{-/-}$ mice, we stained the small intestine for ephrin-B1, ephrin-B2, and EphB2. Localization of ephrin-B1 and ephrin-B2 decreased gradually toward the bottom of WT crypts (Fig 5A and B). This gradient and intensity were severely changed in M3$^{-/-}$ mice. Cells staining strongly for ephrin-B1 and ephrin-B2 were present throughout the crypts (Fig 5A and B). Conversely, EphB2 was expressed throughout the proliferative compartment, peaking at positions 4–6 near the base, and its expression decreased in a gradient toward the top of the crypts in WT mice (Fig 5C). Compared with WT crypts, EphB2 staining in the small intestine of M3$^{-/-}$ mice was increased toward the top of the crypts (Fig 5C). No immuno-histochemical staining was observed in cells in the absence of the primary antibody as a control (Fig 5A–C). Therefore, progenitor cells (transit-amplifying cells) co-expressing EphB2 receptors, and their ligands (ephrin-B1 and ephrin-B2) were increased in a position-dependent pattern in M3$^{-/-}$ crypts.

To determine whether the expression of components of the EphB/ephrin-B system are altered in M3$^{-/-}$ mice, we performed qRT-PCR in the crypts. Gene expression analysis showed that the examined components of the EphB/ephrin-B system in the crypts, with the exception of EphB1 and ephrin-B3, were more highly expressed compared with that of WT crypts (Fig 5D). In particular, the levels of ephrin-B1, ephrin-B2, EphB2, and EphB4 were strongly enhanced compared with the WT control (Fig 5D). Paneth cells preferentially express EphB3 (Holmberg et al, 2006). The increased level of EphB3 coincides with the increase in lysozyme expression (Figs 3E and 5D). Compared with the WT controls, expression of EphB1 was significantly reduced in M3$^{-/-}$ mice, although no change in ephrin-B3 expression was observed between the two groups (Fig 5D). The data in Fig 5A–D suggest that the change in expression of components of the EphB/ephrin-B system affects the proliferation and differentiation of ISCs in M3$^{-/-}$ crypts.

## Single-cell organoid growth and differentiation is also enhanced in M3$^{-/-}$ mice

Next, a monoclonal antibody directed against the extracellular domain of EphB2 was used to label crypt cells (Mao et al, 2004), and fluorescence activated cell sorting (FACS) was used to isolate crypt cells displaying different EphB2 surface expression in WT and M3$^{-/-}$ mice. According to the cell isolation method, we divided into four groups (EphB2$^{high}$, EphB2$^{med}$, EphB2$^{low}$, and EphB2$^{neg}$ cells) (Fig 6A). We then investigated whether EphB2 expression could distinguish among four groups in M3$^{-/-}$ crypts as it carried out in the WT crypts (Fig 6A). Analysis of marker genes revealed similar or higher levels of *Ki67*, *Myc*, and *FoxM1* in EphB2$^{high}$ cells compared with their expression in EphB2$^{med}$ cells in WT and M3$^{-/-}$ crypts (Fig 6B and C). However, both EphB2$^{high}$ cells sorted from WT and M3$^{-/-}$ crypts expressed higher levels of the ISC-specific marker genes, *Lgr5*, *Ascl2*, and *Olfm4* (Fig 6B and C). On the contrary, the levels of these ISC-specific marker genes in EphB2$^{low}$ and EphB2$^{neg}$ cells was extremely low compared with that of EphB2$^{high}$ and EphB2$^{med}$ cells (Fig 6B and C). These results suggest that the EphB2$^{high}$ cell

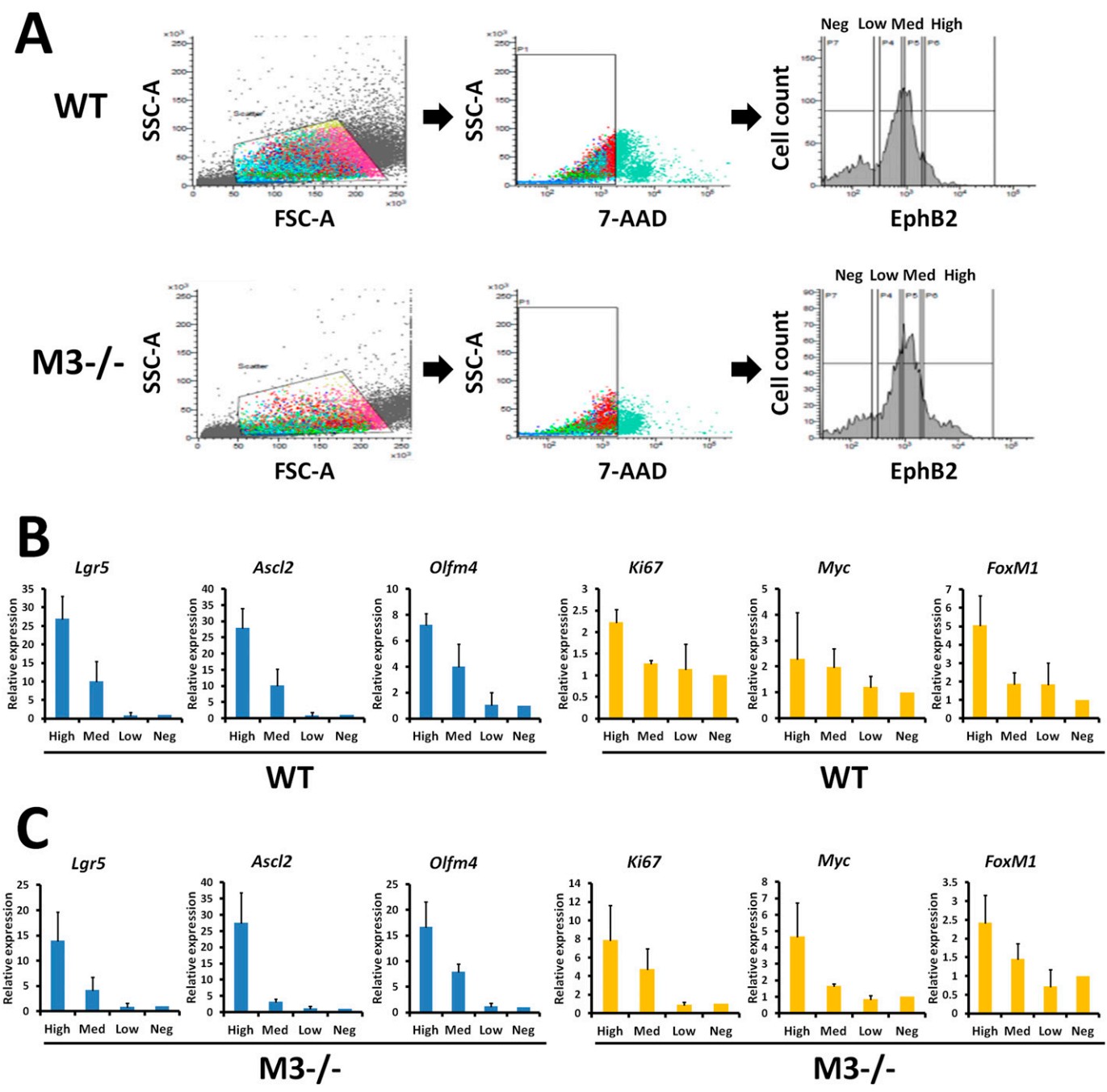

**Figure 6. Isolation of mouse intestinal epithelial cell populations according to EphB2 expression.**
**(A)** Flow cytometry of crypt cells stained for EphB2 to divide into four distinct populations displaying different levels of cell surface EphB2 (high, medium, low, and negative) in WT and M3$^{-/-}$ mice. **(B)** qRT-PCR analysis of stem cell and proliferation marker genes in WT crypts. **(C)** qRT-PCR analysis of stem cell and proliferation marker genes in M3$^{-/-}$ crypts. Data information: The results of flow cytometry are based on six independent experiments. The results of qRT-PCR are based on three independent experiments.
Source data are available online for this figure.

population is enriched in ISCs, whereas EphB2$^{med}$ cells correspond mainly to progenitor cells (transit-amplifying cells) in crypts.

Single cells expressing high levels of EphB2 were sorted by FACS and directly subjected to culture for organoid growth (Fig 7A). These in vitro organoids recreated the organization of the intestinal epithelium, including the presence of multiple crypt-like structures distributed around a central lumen (Fig 7A). Organoid growth of EphB2$^{high}$ cells derived from M3$^{-/-}$ mice was clearly enhanced compared with that of EphB2$^{high}$ cells derived from WT mice (Fig 7A). The enhancement of organoid growth occurred over the course of 9 d (Fig 7A). Except for day 1 and 2, statistically significant enhancement of organoid growth was present until day 9 (Fig 7B). Next,

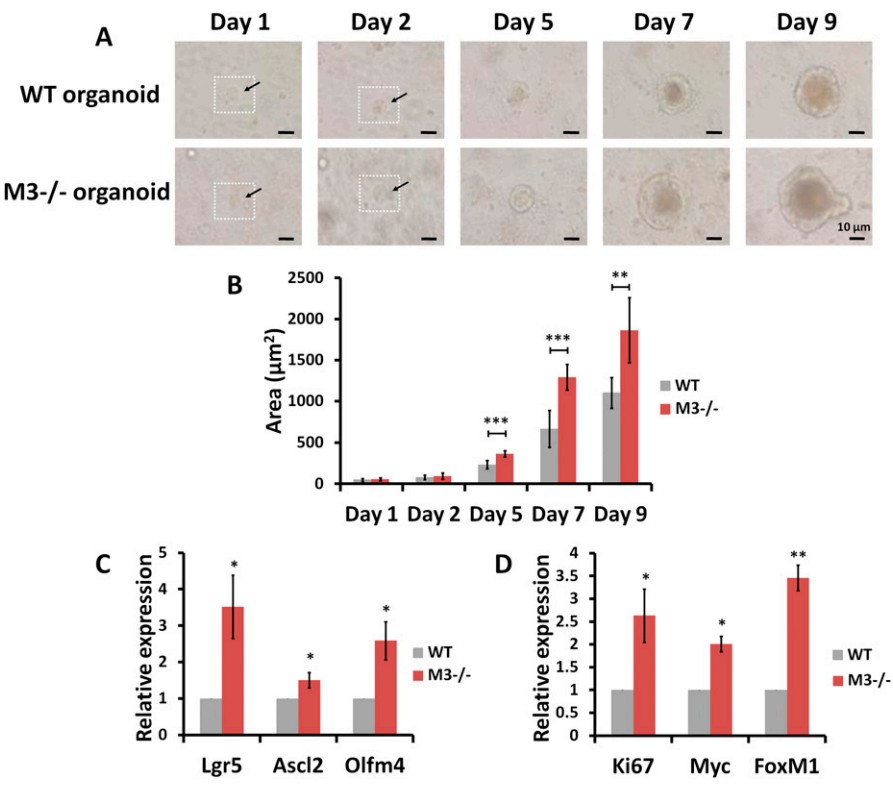

**Figure 7. Comparison of stem and progenitor cell proliferation according to EphB2 expression in WT and M3$^{-/-}$ mice.**
**(A)** Representative phase contrast microscopy images taken at different time points describe organoid growth from single-sorted EphB2$^{high}$ cells in WT and M3$^{-/-}$ mice. **(B)** Comparison of organoid growth from single-sorted EphB2$^{high}$ cells in WT and M3$^{-/-}$ mice. **(C)** Relative quantification of stem cell marker genes in single-sorted EphB2$^{high}$ cells derived from M3$^{-/-}$ mice. **(D)** Relative quantification of progenitor cell marker genes in single-sorted EphB2$^{med}$ cells derived from M3$^{-/-}$ mice. Data information: The results are based on three independent experiments and are presented as mean values ± SD. The statistical significance was calculated with $t$ test (*$P < 0.05$; **$P < 0.005$; ***$P < 0.0005$). *GAPDH* was used as the internal control. Source data are available online for this figure.

using qRT-PCR, we compared the expression levels of selected marker genes (*Lgr5*, *Ascl2*, and *Olfm4* for ISCs; and *Ki67*, *Myc*, and *FoxM1* for progenitor cells) in EphB2$^{high}$ and EphB2$^{med}$ cells derived from WT and M3$^{-/-}$ crypts, respectively. These analyses revealed that the levels of all marker gene transcripts in EphB2$^{high}$ and EphB2$^{med}$ cells derived from M3$^{-/-}$ crypts were significantly up-regulated compared with those derived from WT crypts (Fig 7C and D). Overall, these observations support the idea that the increase in crypt size in M3$^{-/-}$ mice results from higher proliferation rates of ISCs and progenitor cells because of activation of the EphB/ephrin-B system.

### Deletion of M3 influences the activation of the MAPK/ extracellular signal-regulated kinase (ERK) signaling pathway

EphB2 correlates with activity of the MAPK/ERK kinase (MEK) signaling pathway, which is involved in self-renewal and proliferation of germline stem and progenitor cells (N'Tumba-Byn et al, 2020). Because deletion of M3 from the small intestine results in up-regulation of the MAPK signaling pathway (Fig 4), we examined the consequence of M3 deletion on activation of the MAPK/ERK signaling pathway using a MEK inhibitor (U0126).

Immunofluorescence analysis using an anti-phosphorylated ERK1/2 (pERK1/2) antibody revealed extensive phosphorylation of ERK1/2 in M3$^{-/-}$ organoids compared with that in WT organoids (Fig 8A). After treatment with 2 $\mu$M U0126, the extensive phosphorylation in M3$^{-/-}$ organoids was reduced to the WT level (Fig 8A). A significant difference was found in the number of pERK1/2-positive cells in WT organoids, U0126-treated M3$^{-/-}$ organoids, and M3$^{-/-}$ organoids (Fig 8B). Furthermore, the growth of M3$^{-/-}$ organoids treated with 2 $\mu$M U0126 was reduced to that of WT organoids at day 4 (Fig 8C).

Next, we examined the pharmacological effect of U0126 on proliferation of M3$^{-/-}$ organoids using FACS analysis combined with 5-ethynyl-2'-deoxyuridine (EdU) assay to directly detect proliferating cells. In WT organoids, M3$^{-/-}$ organoids, and U0126-treated M3$^{-/-}$ organoids, 9.5%, 19.96%, and 9.73% of cells were EdU-positive (EdU⁺ cells), respectively (Fig 8D). The difference in EdU⁺ cells between WT and M3$^{-/-}$ organoids was significant (Fig 8E). In 2 $\mu$M U0126-treated M3$^{-/-}$ organoids, the number of EdU⁺ cells was significantly reduced to the WT level (Fig 8F). Thus, we concluded that M3 is a key molecule for maintenance of proliferation of stem and progenitor cells via the MAPK/ERK signaling pathway.

## Discussion

M3 is expressed at relatively high levels not only in the hypothalamus but also in many other brain regions (Levey et al, 1994; Yamada et al, 2001; Oki et al, 2005). Selective deletion of M3 in the mouse brain shows a significant decrease in growth hormone and insulin-like growth factor-1, which leads to greatly reduced longitudinal growth (Gautam et al, 2009). Here, we found that a deficiency in M3 caused an intestinal phenotype, which was an increase in crypt size and ISC proliferation and differentiation, without differences in the size of the small intestine despite the reduction in somatic growth of M3$^{-/-}$ mice.

The enlarged crypt morphology can be explained by a specific increase in the transit-amplifying compartment and by increased numbers of ISCs with a proportionate increase in transit-amplifying and Paneth cells. We identified the major intestinal epithelial cell

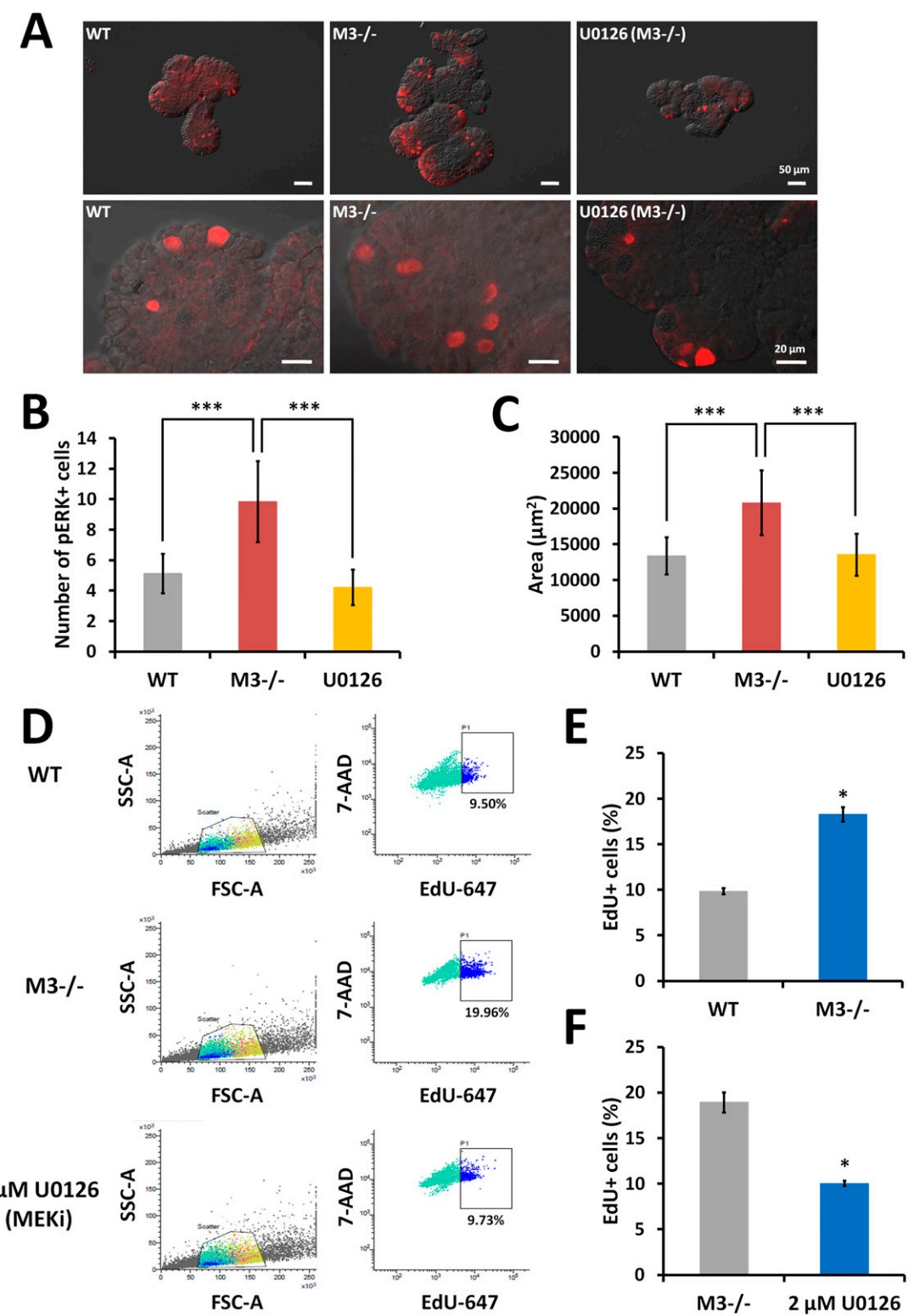

**Figure 8. Pharmacological inhibition of proliferation with a MEK inhibitor (U0126) in M3$^{-/-}$ organoids.**
**(A)** Representative images of anti-pERK antibody staining in WT, M3$^{-/-}$, and U0126-treated M3$^{-/-}$ organoids. **(B)** The number of pERK-positive cells per organoid in WT, M3$^{-/-}$, and U0126-treated M3$^{-/-}$ organoids. At least 25 organoids were examined (n = 3 mice per genotype). **(C)** Comparison of organoid growth in WT, M3$^{-/-}$, and U0126-treated M3$^{-/-}$ organoids at day 4. **(D)** Original FACS data. Cell cycle status of M3$^{-/-}$ small intestinal organoids was determined with flow cytometry. EdU incorporation assay was carried out in WT, M3$^{-/-}$ organoids, and 24 h after addition of a MEK inhibitor (U0126) to regular culture medium in M3$^{-/-}$ organoids. EdU was added 1 h before collection and dissociation of cells for FACS analysis. Single cells were gated using FSC-A/SSC-A characteristics. EdU signals were plotted against 7-AAD signals.

populations according to their differential expression of EphB2 and marker gene expression. Loss of M3 caused profound increase in the population of the EphB2$^{high}$ stem cell and EphB2$^{med}$ progenitor compartment. In addition to their increased populations, we observed increased expression of ISC markers within EphB2$^{high}$ and of progenitor markers within EphB2$^{med}$ populations, supporting an expansion of each compartment upon loss of M3. We also observed increased expression of the marker genes for Paneth cells (*Lysozyme* and *EphB3*) as well as for enterocytes (*villin*) and goblet cells (*mucin-2*) in M3$^{-/-}$ crypts, quantitatively. Paneth cells are the source of multiple stem cell growth factors (e.g., Wnt3, EGF, and TGF-$\alpha$) which are essential signals for stem cell maintenance (Sato et al, 2011). Indeed, co-culturing of sorted ISCs with Paneth cells dramatically improves organoid formation (Sato et al, 2011). Thus, the number of Paneth cell increased can induce to increase the number of ISCs in M3$^{-/-}$ crypts. Collectively, the data support a direct effect of M3 on ISCs, which subsequently affects the size of ISC compartment and the crypt.

Our previous studies using pharmacological tools indicate that the activation of Lgr5$^{+}$ ISCs is most likely to involve the M1, M2, and M3 subtypes (Takahashi et al, 2014). The muscarinic receptors expressed on epithelial cells and/or Lgr5$^{+}$ ISCs may modulate the renewal rate of the intestinal epithelium (Takahashi et al, 2014). Consistent with this observation, M3 contributes to small intestinal mucosal homeostasis (McLean et al, 2016) and is involved in the cholinergic intestinal niche to maintain epithelial homeostasis (Middelhoff et al, 2020). Results of the current study show that deletion of M3 causes an increased rate of proliferation of ISCs and undifferentiated cells and further substantiates the inhibitory effect of M3 on epithelial cell differentiation. These results suggest that one of the physiological functions of M3 in the intestinal tract is to regulate proliferation of ISCs and/or undifferentiated cells and that its deletion results in an increase in the rate of division of these cells. This increase can also explain the increase in migration of epithelial cells along the crypt-villus axis. Our RNA-seq analysis showed that in the ISC niche, M3$^{-/-}$ crypts expressed important genes for migration and proliferation in conjunction with neuronal and/or non-neuronal ACh signaling. The enteric nervous system (ENS) is considered a part of this niche (Sailaja et al, 2016). Mucosal afferent nerves control ISCs and progenitor cells (Bjerknes & Cheng, 2001; Lundgren et al, 2011). In a more recent study, co-culture of a monolayer of organoid-grown differentiated cells with dissociated adult mouse ENS cells shows that the epithelial cell density increases by 40% (Puzan et al, 2018). Especially, chromogranin A–positive epithelial cells increased, suggesting enhancement of enteroendocrine cell population (Puzan et al, 2018). The data imply that epithelial lineage composition may ultimately benefit from the presence of the ENS. As one of the major pathways of excitatory transmission within ENS is mediated by cholinergic transmission (Galligan et al, 2000), neuronal ACh is expected to have the potential functional effects on crypt homeostasis. In the absence of any influence from the ENS, non-neuronal ACh is considered to control the maintenance and differentiation of Lgr5$^{+}$ ISCs in organoids (Takahashi et al, 2014). Ablation of M3 impacts organoid growth through mechanisms such as enhancement of proliferation of Lgr5$^{+}$ ISCs and enhancement of differentiation of enterocytes, goblet cells, and Paneth cells, but not enteroendocrine cells, in the crypt-villus axis. The result agrees with previous studies showing that expression of the gene encoding chromogranin A is not altered after treatment with scopolamine (Middelhoff et al, 2020). We conclude that the function of Lgr5$^{+}$ ISCs is, at least in part, under the control of an independent epithelial cholinergic system mediated by the EphB/ephrin-B system.

Wnt proteins are master regulators of cell proliferation and maintenance in the intestine. EphB2 receptor expression in the intestine is under the control of Wnt proteins, and inhibition of this pathway results in abolished EphB2 expression (Batlle et al, 2002; van de Wetering et al, 2002). The present data demonstrate that M3 signaling directly controls the EphB/ephrin-B system independent of the Wnt signaling pathway. The M3 ligand, ACh, is a soluble small molecule, and the physical range of the effect of ACh may indicate that it is an autocrine/paracrine signal (Takahashi et al, 2014). EphB2 expression extends substantially higher up in the crypt than the domain of cells that express nuclear $\beta$-catenin (Batlle et al, 2002). On the other hand, the expression of ephrin-B1 and B2 is negatively regulated by $\beta$-catenin (Batlle et al, 2002). M3 is expressed in numerous cells at the crypt base as well as in the +4 to +5 cell position (Haber et al, 2017; Middelhoff et al, 2020). Thus, concerning cell movement from the crypt bottom to top, the cells are exposed and maintained under the control of the M3 cholinergic pathway. EphB/ephrin-B signaling activates ERK to promote proliferation in cultured mouse mesenchymal cells (Bush & Soriano, 2010). We revealed that treatment with the ERK1/2 inhibitor, U0126, in M3$^{-/-}$ organoids, resulted in a decrease in the number of EdU$^{+}$ cells and organoid growth to WT levels. These results indicate that MAPK/ERK signaling is downstream of EphB/ephrin-B signaling. Phosphoinositide 3-kinase signaling is activated by the cholinergic receptor blocker, scopolamine, and in M3-conditional knockout mice (Middelhoff et al, 2020). Thus, the three-part mechanism of phosphoinositide 3-kinase, EphB/ephrin-B, and MAPK/ERK signaling occurs in crypts in M3$^{-/-}$ mice.

The evidence that EphB and ephrin-B associate and interact with other cell surface receptors such as channel type receptors and G protein-coupled receptors suggests integration of different signaling routes or modulation of signal transmission. The N-methyl-D-aspartate receptor channel, which is a receptor for glutamate in the central nervous system, interacts with EphB at the cell surface; this interaction is mediated by the extracellular regions of the two receptors (Dalva et al, 2000). Bruno and coworkers (Calò et al, 2005) have provided evidence for a novel type of interaction between ephrin-B2, N-methyl-D-aspartate receptors, and metabotropic glutamate 1 receptors, a new partner in the network in the developing brain. M3 and metabotropic glutamate 1 receptors are both coupled to Gq proteins, and their activation stimulates phosphatidylinositol hydrolysis with ensuing

---

**(E)** Comparison of EdU-positive cells between WT and M3$^{-/-}$ organoids at day 4. **(F)** Comparison of EdU-positive cells between M3$^{-/-}$ and U0126-treated M3$^{-/-}$ organoids at day 4. Data information: Relative percentage of EdU-positive cells is shown for six independent experiments. The result is presented as mean values ± SD. The statistical significance was calculated with *t* test (*$P$ < 0.05). **(B, C)** The results (B, C) are based on three independent experiments and are presented as mean values ± SD. The statistical significance was calculated with *t* test (***$P$ < 0.0005).
Source data are available online for this figure.

intracellular $Ca^{2+}$ release and activation of protein kinase C (De Blasi et al, 2001). This is the first time that M3 signaling has controlled the EphB/ephrin-B system in the small intestine. However, the molecular details of the new link between M3 and EphB/ephrin-B signaling remain unclear.

Identification of signaling pathways that have divergent effects in tissue stem/progenitor and cancer cells may offer insights into cancer development as well as offer novel therapeutic targets. M3 is interesting in this context, because genetic inhibition of M3 activity and treatment with the muscarinic receptor antagonist, scopolamine butyl-bromide, attenuate small intestinal adenoma formation in Apc$^{min/+}$ mice (Raufman et al, 2011). In addition, EphB receptors act as tumor suppressors for colon carcinoma development despite enhancing cell proliferation in the intact intestinal epithelium as well as in adenomas (Batlle et al, 2005; Genander et al, 2009). The tumor suppressor function of EphB receptors is considered to be a result of their regulation of cell migration and compartmentalization of tumor cells (Cortina et al, 2007). Indeed, analysis of Apc$^{min/+}$ epithelium on the EphB4-deficient background revealed that regulation of cell proliferation, extracellular matrix remodeling, and invasive potential are important mechanisms of tumor suppression by EphB4 (Dopeso et al, 2009). We showed that the expression of *EphB4* was strongly increased in M3$^{-/-}$ crypts. Furthermore, the pathway of "chemical carcinogenesis" was down-regulated in the crypts. Thus, muscarinic receptor; antagonists may prove useful in the prevention or treatment of intestinal neoplasia. Furthermore, M3 and post-M3 signaling such as EphB/ephrin-B signaling may be novel therapeutic targets for adenoma growth and potentially cancer progression in the intestinal epithelium.

We conclude that a functional interaction between M3 and the EphB/ephrin-B system plays an important role in the regulation of ISCs in normal adult mouse crypts. EphB2 becomes gradually silenced as cells differentiate from the bottom to the top in WT crypts, whereas ephrin-B ligand-positive precursors are organized from the top to the bottom according to a reverse gradient of ephrin-B1 and ephrin-B2 expression (Fig 9). Upon deletion of M3, suppression by M3 with regards to the EphB/ephrin-B system is removed, and proliferation and positioning in the progenitor domain at the side of the crypt is then increased (Fig 9). Thus, expression of M3 enables ISCs and progenitor cells to fine-tune their cellular response upon ACh stimulation and ensures maintenance of intestinal tissue homeostasis.

## Materials and Methods

### Animals

The generation of M3$^{-/-}$ mice has been described previously (Yamada et al, 2001). Corresponding WT mice (C57BL/6 obtained from Charles River) were used as controls. Both male and female mice were used. This study was approved by the Suntory animal ethics committee (APRV000561), and all animals were maintained in accordance with committee guidelines for the care and use of laboratory animals. Mice were euthanized with $CO_2$ asphyxiation.

### Crypt isolation and crypt–villus organoid culture

Crypt isolation and crypt-villus organoid culture were performed (Takahashi et al, 2014). Briefly, proximal parts of the small intestine

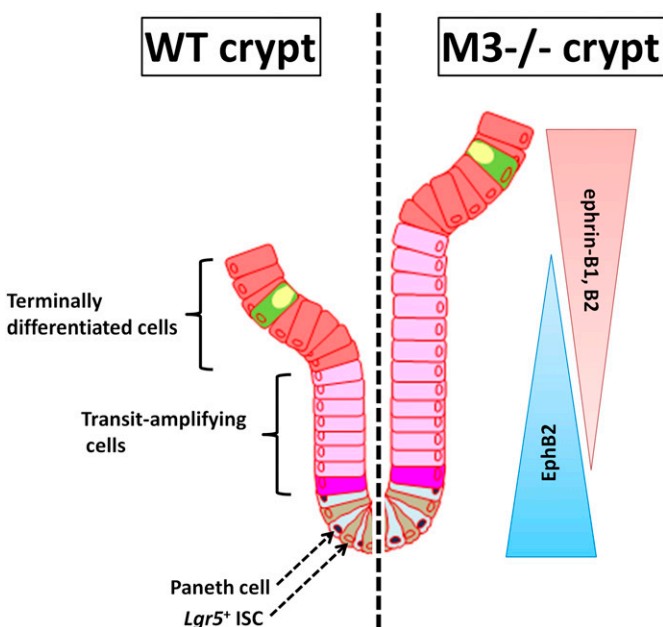

**Figure 9. Model depicting the proposed role of M3 through EphB/ephrin-B signaling in the intestinal stem cell niche.**
Disruption of M3 signaling alters the number of intestinal stem cells and progenitor cells through activation of EphB/ephrin-B and eventually increases the size of the proliferative compartment.

were opened longitudinally and washed with cold PBS including penicillin/streptomycin (10 mg/ml) and gentamicin sulfate solution (10 mg/ml) (PBS-ABx). The tissue was cut into 5 × 5-mm pieces that were further washed with cold PBS-ABx. The washed tissue fragments were incubated in PBS-ABx containing 2 mM EDTA for 30 min on ice. After removal of the EDTA solution, the tissue fragments were vigorously shaken in cold PBS-ABx. The resulting suspension was passed through a 70-µm cell strainer (BD Biosciences) to remove residual villous material, and the isolated crypts were then centrifuged (390*g*, 3 min, 4°C). The final fraction consisted of essentially pure crypts, and was used for culture. Isolated crypts were counted and pelleted by centrifugation at 290*g* for 3 min at 4°C. A total of 100 crypts were mixed with 40 µl Matrigel (BD Biosciences) and plated in 24-well plates pre-warmed to 37°C. After Matrigel polymerization, 500 µl crypt culture medium (Advanced Dulbecco's Modified Eagle's Medium/F12; Gibco) containing growth factors (20 ng/ml mouse epidermal growth factor [R&D Systems], 500 ng/ml R-spondin 1 [R&D Systems], and 100 ng/ml Noggin [R&D Systems]) was added. The organoids were maintained at 37°C in a humidified atmosphere with 5% carbon dioxide. Culture medium was changed every other day. Crypt size and organoid surface area were analyzed with the Image J program (NIH).

### Histology and immunohistochemistry

The small intestines were removed from WT and M3$^{-/-}$ mice for histological and immunohistochemical characterization of the tract. The small intestines were divided into three parts (duodenum, jejunum, and ileum), and each part was fixed in Bouin's fixative and embedded in paraffin. Ten-micrometer-thick sections

were cut with a microtome (IVS-410; SAKURA) and stained with the routine H&E method.

Some sections were used for immunohistochemical staining. After antigen retrieval, sections were deparaffinized in xylene, incubated in 0.3% hydrogen peroxide in methanol for 30 min, and incubated in blocking buffer (3% *bovine serum albumin* in PBS) for 1 h at room temperature. Sections were then incubated with goat anti-EphB2 (1:50; R&D Systems), goat anti-ephrin-B1 (1:50; R&D Systems), and rabbit anti-ephrin-B2 (1:50; R&D Systems) overnight at 4°C. Detection of primary antibodies for immunohistochemistry was carried out using an appropriate secondary antibody at 1:250 dilution for 1 h at room temperature; 3,3′-diaminobenzidine solution was used for detection and visualization of staining. This was followed by washes with PBS and mounting with MOUNT-QUICK (DAIDO SANGYO). Control sections were processed in the absence of the primary antibody. Nuclei were labeled with hematoxylin.

### Immunofluorescence

Intestinal tissue samples were embedded in Tissue-Tek OCT compound (QIAGEN), frozen at −20°C, and then cut serially into 10-$\mu$m-thick sections with a cryostat (Thermo Fisher Scientific). For immunofluorescence, the sections were fixed with 4% paraformaldehyde (Nacalai Tesque) for 10 min at room temperature. After three washes with PBS, the sections were treated with 0.1% sodium borohydride (Nacalai Tesque) for 10 min at room temperature to reduce autofluorescence, and were again washed three times with PBS. The sections were then treated with 0.1% Triton X-100 (Nacalai Tesque) for 10 min at room temperature, and washed with PBS three times. The sections were incubated with blocking solution containing 1% bovine serum albumin (Sigma-Aldrich) for 10 min at room temperature followed by incubation for 60 min at room temperature in rabbit anti-Ki67 antibody (1:500, Cat. No. 16667; Abcam) diluted in blocking solution. After incubation, the sections were washed three times with PBS and further incubated with Alexa Fluor 488 goat anti-rabbit IgG (1:1,000; Thermo Fisher Scientific). Nuclei were stained with Hoechst 33342 (1:1,000; AnaSpec). Sections were mounted in Mowiol mounting medium (Mowiol 4-88; Sigma-Aldrich) under a cover glass, and observed by confocal immuno-fluorescence microscopy (FV1000; Olympus). Following immuno-histochemical staining for Ki67, the number of Ki67-positive cells was counted from at least 30 crypts of WT and M3$^{-/-}$ mice.

For whole-mount staining of organoids, they were collected in ice-cold medium, pelleted, and resuspended in chilled Cell Recovery Solution (BD). After incubation (15 min on ice) and gentle mixing, organoids were washed in TBS, pelleted, resuspended, and fixed in 4% paraformaldehyde overnight at 4°C. Organoids were subsequently permeabilized in 0.1% Tween20 (30 min at room temperature) before resuspension in blocking solution at 4°C for 1 h. After incubation in anti-pERK1/2 antibody (#9101, RRID: AB_331646; Cell Signaling Technology) (1:200, 4°C overnight), three washing steps were performed at room temperature by addition of TBS and organoid sedimentation. Organoids were then incubated in the secondary antibody (Alexa Fluor 546 goat anti-rabbit IgG, 1:1,000) (Molecular Probes). Control organoids were incubated with pre-absorbed primary antibody. Organoids were mounted in Mowiol mounting medium under cover glass, and observed by FV1000 confocal immunofluorescence microscopy.

### BrdU labeling

WT and M3$^{-/-}$ mice were injected intraperitoneally with BrdU (Nacalai Tesque) at 50 mg/kg body weight as a marker of proliferation, and then euthanized at 3, 6, 15, and 24 h post-injection. After immunostaining, BrdU labeling was detected using the BrdU detection and labeling kit II (Boehringer-Mannheim) according to the manufacturer's instructions. Following immunohistochemical staining for BrdU, the number of BrdU-positive cells was counted from at least 30 crypts of WT and M3$^{-/-}$ mice per time point.

### RT–PCR and qRT-PCR

Total RNAs from crypts, villi, and organoids were extracted with TRIzol reagent (Gibco) according to the manufacturer's instructions. Intestinal villi from WT and M3$^{-/-}$ mice were scraped off with glass slides. The villi collected were washed with PBS and filtered with a 70-$\mu$m cell strainer to prevent crypt contamination. The extracted RNAs were treated with DNase I (Ambion) to eliminate genomic DNA from RNA preparations. Total RNA (3 $\mu$g) was used as a template for cDNA synthesis. Reverse transcription was performed with SuperScript II and an oligo-dT primer according to the protocol recommended by the manufacturer (Invitrogen). cDNAs encoding mAChR subtypes (M1-M5) were amplified with PCR with designed PCR primers (Table 1). RT–PCR was carried out using a GeneAmp PCR System 9700 (Applied Biosystems) with conditions as follows: initial denaturation at 94°C for 3 min; followed by 35 cycles at 94°C for 30 s, 55°C for 30 s, and 72°C for 1 min, and then one final extension step at 72°C for 5 min. The expression of glyceraldehyde 3-phosphate dehydrogenase (*GAPDH*) was used as an internal control.

Specific gene expression levels were analyzed with qRT-PCR. Isolation of total RNA and cDNA synthesis were performed as described above. qRT-PCR for specific genes was performed in triplicate using SYBR Green master mixture (Bio-Rad), according to the manufacturer's instructions. qRT-PCR was carried out on a CFX96 Real-Time System (Bio-Rad) with conditions as follows: polymerase activation and DNA denaturation for 30 s at 95°C, followed by 45 cycles at 95°C for 10 s and 55°C for 30 s; then 65°C for 5 s, and then 95°C for 50 s for melt-curve analysis. *GAPDH* was amplified as an internal control. All primers for qRT-PCR are presented in Table 2. For relative quantification of gene expression, the comparative C$_T$ method was used.

### RNA-seq

Total RNAs from WT and M3$^{-/-}$ crypts were extracted from log phase cultures using an RNeasy Mini Kit (QIAGEN). The quality of the RNA samples was evaluated using a BioAnalyzer (Agilent Technologies) with the RNA6000 Nano Chip. Total RNA (2 $\mu$g) from each sample was used to construct cDNA libraries using TruSeq Stranded mRNA (Illumina) according to the manufacturer's instructions. The cDNA libraries were validated using the BioAnalyzer with the DNA1000 Chip and quantified using the Cycleave PCR Quantification Kit (Takara Bio). Single end sequencing using 101 cycles was performed

**Table 1.   Primers for RT–PCR.**

| Target | Forward primer | Reverse primer |
|---|---|---|
| M1 | 5'-TCCCTCACATCCTCCGAAGGTG-3' | 5'-CTTTCTTGGTGGGCCTCTTGACTG-3' |
| M2 | 5'-GGACCTGTAGTGTGCGACCT-3' | 5'-CCCGTCTTCCACAGTCCTTA-3' |
| M3 | 5'-GCTCAGAGACCAGAGCCATC-3' | 5'-ACAGTTGTCACGGTCATCCA-3' |
| M4 | 5'-TCCTCACCTGGACACCCTAC-3' | 5'-TTGAAAGTGGCATTGCAGAG-3' |
| M5 | 5'-TCAGCCATCAAATGACCAAA-3' | 5'-AGTAACCCAAGTGCCACAGG-3' |
| GAPDH | 5'-AACTTTGGCATTGTGGAAGG-3' | 5'-ACACATTGGGGGTAGGAACA-3' |

using HiSeq1500 (Illumina) in the rapid output mode. Total reads were extracted with CASAVA v1.8.2 (Illumina). PCR duplications, adaptor sequences, and low quality reads were removed from the extracted reads as follows. If the first 10 bases of the two reads were identical and the entire reads showed >90% similarity, the reads were considered to be PCR duplicates. Base calling from the 5' to the 3' end was stopped when the frequency of accurately called bases dropped to 0.5. The NextGen sequencing data were deposited in the Sequence Read Archive database (accession numbers: SRR12489254–SRR12489266). The remaining reads were aligned to mouse reference transcripts (build mm10) from the University of California Santa Cruz genome browser database. The alignment parameters were set to allow a single mutation in alignment and ignore the mismatch penalty for low quality nucleotides with lower quality than 20. The alignment was performed against the mouse reference genome using Cufflinks (v2.0.10). The reference genome and annotations were downloaded in fasta and gene transfer format from the University of California Santa Cruz genome browser database. All parameters for Cufflinks were set to default values. DEGs were selected if they showed a greater than twofold change between the two groups and were subsequently assessed with DAVID (Dennis et al, 2003). From DAVID analysis, DEGs were

**Table 2.   Primers for quantitative RT–PCR.**

| Target | Forward primer | Reverse primer |
|---|---|---|
| M1 | 5'-AGCAGCAGCTCAGAGAGGTC-3' | 5'-GCCTGTGCCTCAGGATCTAC-3' |
| M2 | 5'-CGGCTTTCTATCTGCCTGTC-3' | 5'-GGCATGTTGTTGTTGTTTGG-3' |
| M3 | 5'-GCTCAGAGACCAGAGCCATC-3' | 5'-ACAGTTGTCACGGTCATCCA-3' |
| M4 | 5'-TCCTCACCTGGACACCCTAC-3' | 5'-TTGAAAGTGGCATTGCAGAG-3' |
| M5 | 5'-TCAGCCATCAAATGACCAAA-3' | 5'-AGTAACCCAAGTGCCACAGG-3' |
| Lgr5 | 5'-AGCATACCCGTTTCTGGATG-3' | 5'-AGGACCGTTTCTCAACATCG-3' |
| Lys | 5'-ATGGCTACCGTGGTGTCAAG-3' | 5'-GGGATCTCTCACCACCCTCT-3' |
| Vill | 5'-CTTCTTCGATGGTGACTGCTAT-3' | 5'-AAGTCTCGCTCTCGTTGCCT-3' |
| Muc2 | 5'-TGCCCAGAGAGTTTGGAGAGG-3' | 5'-CCTCACATGTGGTCTGGTTG-3' |
| ChgA | 5'-CAGGCTACAAAGCGATCCAG-3' | 5'-GCCTCTGTCTTTCCATCTCC-3' |
| ephrin-B1 | 5'-TCGCAAGCATACACAGCAGCGG-3' | 5'-ATGATGATGTCGCTGGGCTCGG-3' |
| ephrin-B2 | 5'-CAGAAGAACCCTGCTTGCCTGG-3' | 5'-AGCAAGCAGCCTTGACCTGC-3' |
| ephrin-B3 | 5'-AGACTTTGGGGGAGTTGGTGCC-3' | 5'-CAGCCCCGCAAAACCTAACAGC-3' |
| EphB1 | 5'-TTACAGCACAGGCCGAGGGGAGTTCG-3' | 5'-AACTGGCCCATGATGCTGCC-3' |
| EphB2 | 5'-ACGCCACGGCCATAAAAAGCCC-3' | 5'-TTGCCACTGTAGCGCCCATAGC-3' |
| EphB3 | 5'-ATTGGGCATCAAGCCACCCAGC-3' | 5'-TGCTCTGTAACCGAGGTGTCGC-3' |
| EphB4 | 5'-TTGAGCCCTGGGTGGCAATCCG-3' | 5'-AGGCACCTCACGGTCAGTGG-3' |
| EphB6 | 5'-ACTCTAAGCTGCGAGCAGACGC-3' | 5'-GCCAGGCTTGCCTTCTTGTCTGG-3' |
| Ascl2 | 5'-TCCTGGTGGACCTACCTGCTT-3' | 5'-AGGTCAGTCAGCACTTGGCATT-3' |
| Olfm4 | 5'-CAGCCACTTTCCAATTTCACTG-3' | 5'-GCTGGACATACTCCTTCACCTTA-3' |
| Ki67 | 5'-AGTCTCTTGGCACTCACAGC-3' | 5'-ATTTTGTAGGGTCGGGCAGG-3' |
| Myc | 5'-AGTGCATTGACCCCCTCAGTG-3' | 5'-TCAGCTCGTTCCTCCTCTGA-3' |
| FoxM1 | 5'-AGCCTGAGGAGGACATAGCA-3' | 5'-GGGTTCGTACTGGGCTGAAA-3' |
| GAPDH | 5'-TGACGTGCCGCCTGGAGAAA-3' | 5'-AGTGTAGCCCAAGATGCCCTTCAG-3' |

categorized into KEGG pathways. Enriched pathways were selected from KEGG categories that had more than five up-regulated and down-regulated genes ($1 \times 10^{-3}$), and annotated as up-regulated and down-regulated pathways in M3$^{-/-}$ crypts.

## Small intestinal crypt cell purification and single-cell cultured organoids

Isolation of small intestinal crypts was performed as described above in the "Crypt isolation and crypt-villus organoid culture" section. Isolated crypts were then enzymatically disaggregated (1 ml TrypLE Express; Gibco) for 30 min at 37°C with orbital shaking to obtain single-cell suspensions. After disaggregation, 5% fetal bovine serum was added, and cells were passed through a 20-$\mu$m cell strainer (Celltrix; SYSMEX) and washed with PBS. Cells were then centrifuged (390$g$ for 5 min at 4°C) and resuspended in staining buffer (5% fetal bovine serum in Hanks' Balanced Salt Solution) for 5 min. Anti-EphB2 APC-conjugated antibody (1:50, clone 2H9; BD Biosciences) was then added and incubated for 30 min on ice. Cells were then washed twice with PBS. Finally, 7-AAD (BD Biosciences) was added (1:100), and stained cells were sorted in a FACSMelody (BD Life Sciences-Biosciences). To obtain the EphB2-positive (EphB2$^+$) and EphB2-negative cell populations, dead cells and debris were discarded by removing the 7-AAD-positive subpopulation. Different intestinal epithelial cells were selected according to graded EphB2 surface levels. The brightest EphB2$^+$ cells were sorted and divided into three groups, EphB2-high (EphB2$^{hi}$), EphB2-med (EphB2$^{med}$), and EphB2-low (EphB2$^{low}$) groups. The EphB2-negative (EphB2$^{neg}$) subpopulation did not stain for EphB2.

Single-sorted EphB2$^{hi}$ cells were collected, pelleted, and embedded in Matrigel (BD Bioscience), followed by seeding on a 24-well plate (30–50 singlets/40 $\mu$l Matrigel/well). After Matrigel polymerization, culture medium was overlaid. Y-27632 (10 $\mu$M) (FUJIFILM Wako Pure Chemical Corporation) was included for the first 2 d to avoid anoikis. The cells were manually inspected by inverted microscopy (TE2000-S; Nikon), and the numbers of viable organoids in triplicate were calculated.

## Pharmacological assay

A pharmacological assay was performed with 3-d cultured organoids in the presence or absence of a MEK inhibitor (U0126; Promega). U0126 was dissolved in dimethylsulfoxide (Nacalai Tesque). For control organoids, regular medium with dimethylsulfoxide was used to test the effect of U0126. The drug was added directly to 3-d cultures of organoids (time zero), and the organoids were collected for analysis 1 d later. To assay growth, the organoid surface area after 4 d of culture was analyzed with the Image J program (NIH). To detect the effect of U0126 on the cell cycle of the organoids, we carried out immunofluorescence using the anti-pERK1/2 antibody (#9101, RRID: AB_331646; Cell Signaling Technology). Following immunohistochemical staining for pERK1/2, the number of pERK1/2–positive cells was counted from at least 30 organoids of WT and M3$^{-/-}$ mice.

## EdU incorporation assay

The EdU incorporation assay was performed using the Click-it EdU Alexa Fluor 647 Flow Cytometry assay kit (Life Technologies). The assay was carried out in control cultures and after addition of the MEK inhibitor (U0126) to regular culture medium for 24 h. Cells derived from organoids were treated with 10 $\mu$M EdU for 1 h before harvesting and single-cell dispersal (as described above) for FACS analysis. Genomic DNA content was measured by addition of 7-AAD (1:100) before analysis.

## Statistical analysis

Comparisons between two groups of data were made with two-tailed, unpaired $t$ tests. Data and statistical analyses were performed with Microsoft Excel. Data are presented as means ± standard deviation (means ± SD). In all cases, differences were considered to be significant when $P < 0.05$. All experiments were repeated at least three times.

# Data Availability

All data needed to evaluate the conclusions in the article are present in the article and/or the Supplementary Materials. The NextGen sequencing data are openly available in the Sequence Read Archive database (accession numbers: SRR12489254-SRR12489266). Additional data related to this work may be requested from the corresponding author.

# Supplementary Information

# Acknowledgements

We thank Dr Jürgen Wess (National Institute of Diabetes and Digestive and Kidney Diseases) for providing M3$^{-/-}$ mice. This work was supported by Grants-in-Aid for Scientific Research (C) to T Takahashi (grant numbers: JP17K07495 and JP20K06751).

## Author Contributions

T Takahashi: conceptualization, data curation, formal analysis, supervision, funding acquisition, validation, investigation, methodology, project administration, and writing—original draft.
A Shiraishi: software, validation, and methodology.
J Murata: validation, investigation, and methodology.
S Matsubara: validation, investigation, and methodology.
S Nakaoka: validation, investigation, and methodology.
S Kirimoto: validation, investigation, and methodology.
M Osawa: supervision and writing—review and editing.

## Conflict of Interest Statement

The authors declare that they have no conflict of interest.

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
