## [Reviewer comments · Life Science Alliance]

Life Science Alliance

Muscarinic receptor M3 contributes to intestinal stem cell maintenance via EphB/ephrin-B signaling

Akira Shiraishi, Jun Murata, Shin Matsubara, Satsuki Nakaoka, Shinji Kirimoto, Masatake Osawa and Toshio Takahashi

DOI: <https://doi.org/10.26508/lsa.202000962>

Corresponding author(s): Dr. Toshio Takahashi (Suntory Foundation for Life Sciences)

Review Timeline:

Submission Date:	2020-11-17
Editorial Decision:	2021-06-03
Revision Received:	2021-06-22
Editorial Decision:	2021-06-23
Revision Received:	2021-06-30
Accepted:	2021-07-01

Transaction Report:

June 3, 2021

Re: Life Science Alliance manuscript #LSA-2020-00962-T

Dr. Toshio Takahashi
Suntory Foundation for Life Sciences
8-1-1 Seikadai
Seika-cho
Kyoto, Kyoto 619-0284
Japan

Dear Dr. Takahashi,

Thank you for submitting your manuscript entitled "Muscarinic receptor M3 contributes to intestinal stem cell maintenance via EphB/ephrin-B signaling" to Life Science Alliance. The manuscript was assessed by an expert reviewer, whose comments are appended to this letter.

Please address the Reviewer's comments, including Paneth cell quantification and a toning down of the Conclusions.

Thank you for this interesting contribution to Life Science Alliance. We are looking forward to receiving your revised manuscript.

Sincerely,

- A letter addressing the reviewers' comments point by point.
- An editable version of the final text (.DOC or .DOCX) is needed for copyediting (no PDFs).
- High-resolution figure, supplementary figure and video files uploaded as individual files: See our detailed guidelines for preparing your production-ready images, <https://www.life-science-alliance.org/authors>
- Summary blurb (enter in submission system): A short text summarizing in a single sentence the study (max. 200 characters including spaces). This text is used in conjunction with the titles of papers, hence should be informative and complementary to the title and running title. It should describe the context and significance of the findings for a general readership; it should be written in the present tense and refer to the work in the third person. Author names should not be mentioned.

B. MANUSCRIPT ORGANIZATION AND FORMATTING:

Reviewer #1 (Comments to the Authors (Required)):

Well done study, written well. This work expands on this groups previous work demonstrating the endogenous (epithelial) cholinergic system controls homeostasis. Clear data support the M3 receptor inhibits Eph/Ephrin signaling with downstream upregulation of MEK, TNF and cell cycle when M3 is KO. Elegant in vitro and in vivo studies demonstrate the direct effect on ISC and expansion of TA compartment. I have no major critic's. Minor comments; would have expected to see Paneth cell quantification. Would also like to understand the role of the ENS better - limit comment in discussion only specifically given the similar in vitro data without signaling

Reviewer #1 (Comments to the Authors (Required)):

Well done study, written well. This work expands on this groups previous work demonstrating the endogenous (epithelial) cholinergic system controls homeostasis. Clear data support the M3 receptor inhibits Eph/Ephrin signaling with downstream upregulation of MEK, TNF and cell cycle when M3 is KO. Elegant in vitro and in vivo studies demonstrate the direct effect on ISC and expansion of TA compartment. I have no major critic's. Minor comments; would have expected to see Paneth cell quantification. Would also like to understand the role of the ENS better - limit comment in discussion only specifically given the similar in vitro data without signaling.

Thank you very much for your positive comments concerning our manuscript. I would like to answer your minor comments as follows.

Paneth cell quantification

(Answer): Concerning Paneth cell quantification, we carried out quantitative RT-PCR to evaluate the functional consequence of deletion of M3. mRNA level of Paneth cell marker gene (*Lysozyme*) was significantly enhanced in M3^{-/-} crypts (1.5 folds over WT crypts) (Fig. 3E). Furthermore, mRNA level of *EphB3* gene which is mainly expressed in Paneth cells was also enhanced compared with that of WT crypts (2 folds) (Fig. 5D). The data support increased numbers of ISCs with a proportionate increase in Paneth cells.

To have a fruitful discussion in our manuscript, I added a new paragraph as follows.

(New paragraph): The enlarged crypt morphology can be explained by a specific increase in the transit-amplifying compartment and by increased numbers of ISCs with a proportionate increase in transit-amplifying and Paneth cells. We identified the major intestinal epithelial cell populations according to their differential expression of EphB2 and marker gene expression. Loss of M3 caused profound increase in the population of the EphB2^{high} stem cell and EphB2^{med} progenitor compartment. In addition to their increased populations, we observed increased expression of ISC markers within EphB2^{high} and of progenitor markers within EphB2^{med} populations, supporting an expansion of each compartment upon loss of M3. We also observed increased expression of the marker genes for Paneth cells (*Lysozyme* and *EphB3*) as well as for enterocytes (*villin*) and goblet cells (*mucin-2*) in M3^{-/-} crypts, quantitatively. Paneth cells are the source of multiple stem cell growth factors (for example, Wnt3, EGF, and TGF- α) which are essential signals for stem cell maintenance (Sato *et al*, 2011). Indeed, co-culturing of sorted ISCs with Paneth cells dramatically improves organoid formation (Sato *et al*, 2011). Thus, the number of Paneth cell increased can induce to increase the number of ISCs in M3^{-/-} crypts. Collectively, the data support a direct effect of M3 on ISCs, which subsequently affects the size of ISC compartment and the crypt.

The role of the ENS

I agree with the reviewer's comment. Understanding the role of the ENS in the intestinal crypts is of key importance. According to the reviewer's comment, I added new sentences in the third paragraph in discussion and new references as follows.

(New sentences): In a more recent study, co-culture of a monolayer of organoid-grown differentiated cells with dissociated adult mouse ENS cells shows that the epithelial cell density increases by 40% (Puzan *et al*, 2018). Especially, chromogranin A-positive epithelial cells increased, suggesting enhancement of enteroendocrine cell population (Puzan *et al*. 2018). The data imply that epithelial lineage composition may ultimately benefit from the presence of the ENS. As one of the major pathways of excitatory transmission within ENS is mediated by cholinergic transmission (Galligan *et al*, 2000), neuronal ACh is expected to have the potential functional effects on crypt homeostasis.

(New references)

1) Puzan M, Hosic S, Ghio C, Koppes A (2018) Enteric nervous system regulation of intestinal stem cell differentiation and epithelial monolayer function. *Sci Rep* 8: 6313

2) Galligan JJ, LePard KJ, Schneider DA, Zhou X (2000) Multiple mechanisms of fast excitatory synaptic transmission in the enteric nervous system. *J Auton Nerv Syst* 81: 97-103

The new link between M3 and EphB/ephrin-B signaling

To have a fruitful discussion in our manuscript, I added a new paragraph as follows. Additionally, I added new references as follows.

(New paragraph): The evidence that EphB and ephrin-B associate and interact with other cell surface receptors such as channel type receptors and G protein-coupled receptors suggests integration of different signaling routes or modulation of signal transmission. The N-methyl-D-aspartate (NMDA) receptor channel, which is a receptor for glutamate in the central nervous system, interacts with EphB at the cell surface; this interaction is mediated by the extracellular regions of the two receptors (Dalva *et al*, 2000). Bruno and coworkers (Bruno *et al*, 2005) have provided evidence for a novel type of interaction between ephrin-B2, NMDA receptors, and metabotropic glutamate 1 receptors, a new partner in the network in the developing brain. M3 and metabotropic glutamate 1 receptors are both coupled to Gq proteins, and their activation stimulates phosphatidylinositol hydrolysis with ensuing intracellular Ca²⁺ release and activation of protein kinase C (De Blasi *et al*, 2001). This is the first time that M3 signaling has controlled the EphB/ephrin-B system in the small intestine. However, the molecular details of the new link between M3 and EphB/ephrin-B signaling remain unclear.

(New references)

1) Dalva MB, Takasu MA, Lin MZ, Shamah SM, Hu L, Gale NW, Greenberg ME (2000) EphB receptors interact with NMDA receptors and regulate excitatory synapse formation. *Cell* 103: 945-956

2) Bruno LC, Spinsanti P, Molinari G, Korkhov V, Esposito Z, Patanè M, Melchiorri D, Freissmuth M, Nicoletti F (2005) Interactions between ephrin-B and metabotropic glutamate 1 receptors in brain tissue and cultured neurons. *J Neurosci* 25: 2245-2254

3) De Blasi A, Conn PJ, Pin J, Nicoletti F (2001) Molecular determinants of metabotropic glutamate receptor signaling. *Trends Pharmacol Sci* 22: 114-120

June 23, 2021

RE: Life Science Alliance Manuscript #LSA-2020-00962-TR

Dear Dr. Takahashi,

Thank you for submitting your revised manuscript entitled "Muscarinic receptor M3 contributes to intestinal stem cell maintenance via EphB/ephrin-B signaling". We would be happy to publish your paper in Life Science Alliance pending final revisions necessary to meet our formatting guidelines.

- please use the [10 author names, et al.] format in your references (i.e. limit the author names to the first 10)
- LSA allows supplementary figures, but no EV Figures; please update your callouts for the Supplementary Figures in the manuscript Fig EV1A=Fig S1A; while supplementary figures use the system supplementary Fig S1;
- please add your table legends to the main manuscript text after the main and supplementary figure legends
- please add callouts for Figures 4C and from S3-S10 A and B to your main manuscript text
- please add a Data Availability Statement, including the deposited NextGen sequencing data mentioned in the Materials and Methods.

A. FINAL FILES:

-- Summary blurb (enter in submission system): A short text summarizing in a single sentence the study (max. 200 characters including spaces). This text is used in conjunction with the titles of papers, hence should be informative and complementary to the title. It should describe the context and significance of the findings for a general readership; it should be written in the present tense

and refer to the work in the third person. Author names should not be mentioned.

B. MANUSCRIPT ORGANIZATION AND FORMATTING:

Sincerely,

July 1, 2021

RE: Life Science Alliance Manuscript #LSA-2020-00962-TRR

Author information redacted

Dear Dr. Takahashi,

Thank you for submitting your Research Article entitled "Muscarinic receptor M3 contributes to intestinal stem cell maintenance via EphB/ephrin-B signaling". It is a pleasure to let you know that your manuscript is now accepted for publication in Life Science Alliance. Congratulations on this interesting work.

DISTRIBUTION OF MATERIALS:

Again, congratulations on a very nice paper. I hope you found the review process to be constructive and are pleased with how the manuscript was handled editorially. We look forward to future exciting submissions from your lab.

Sincerely,

Eric Sawey, PhD
Executive Editor
Life Science Alliance